# Screening of *Amaranthus* sp. Varieties for Resistance to Bacterial Wilt Caused by *Ralstonia solanacearum*

**Rachidatou Sikirou** [1,*], **Marie Epiphane Dossoumou** [1], **Judith Honfoga** [2], **Victor Afari-Sefa** [2], **Ramasamy Srinivasan** [3], **Mathews Paret** [4] **and Wubetu Bihon** [5]

1    Laboratoire de Défense des Cultures (INRAB/LDC), Institut National des Recherches Agricoles du Bénin, Cotonou 01 BP 884, Benin; akinnidossoumou@hotmail.fr
2    World Vegetable Center, West and Central Africa—Coastal and Humid Regions, IITA-Benin Campus, Cotonou 08 BP 0932, Benin; judith.honfoga@worldveg.org (J.H.); victor.afari-sefa@worldveg.org (V.A.-S.)
3    World Vegetable Center, Shanhua, Tainan 74151, Taiwan; srini.ramasamy@worldveg.org
4    North Florida Research and Education Center, University of Florida, Quincy, FL 32351, USA; paret@ufl.edu
5    World Vegetable Center, Eastern and Southern Africa, ILRI Campus, Addis Ababa P.O. Box 5689, Ethiopia; wubetu.legesse@worldveg.org
*    Correspondence: rachidatous@yahoo.fr; Tel.: +229-9788-2620

**Abstract:** Bacterial wilt, caused by *Ralstonia solanacearum*, is an emerging constraint in amaranth production in Benin. Host resistance is the most sustainable disease control measure. Ten amaranth varieties including A2002, Bresil (B) -Sel, Madiira 2, AC-NL, GARE ES13-7, Madiira 1, UG-AMES13-2, AM-NKGN, IP-5-Sel and a local variety from Benin were screened for resistance to bacterial wilt. The study was conducted in a screen house and in the naturally contaminated open field during a consecutive rainy and dry season using a randomized complete block design with four and three replications, respectively. In the screen house, plants were inoculated by drenching a 40 mL of bacterial suspension containing $10^8$ CFU/mL of *R. solanacearum* strain NCBI 5 GenBank N° MH397250 at the collar region. The bacterial wilt incidence (BWI) and the area under the disease progress curve (AUDPC) suggested differential reactions of amaranth varieties to the pathogen. BWI and AUDPC were low for UG-AMES13-2, moderate for Madiira 2, AM-NKGN and the local variety and very high for A2002, Bresil (B) -Sel, AC-NL, GARE ES13-7, Madiira 1 and IP-5-Sel. The World Vegetable Center's UG-AMES13-2 can be considered as first choice, which is resistant to *R. solanacearum*, and should be scaled up for seed production towards supporting farmers.

**Keywords:** bacterial wilt; *Ralstonia solanacearum*; amaranth; screening; resistance



## 1. Introduction

The Republic of Benin is experiencing a progressive demographic growth especially in urban areas. The rate of urbanization increased from 38.9% in 2002 to 44.6% in 2013 [1]. This increasing rate of urbanization implies an increasing demand for vegetables, and constitutes an opportunity for the development of the vegetable subsector with respect to production. Thus, many vegetable production sites are being created in periurban areas to meet the growing demand as a complement to supply from rural production zones [2,3]. These emerging production sites are important and provide vegetables to urban areas, and help meet their food preferences and demand [4,5].

Most nutrition, food security and poverty alleviation programs promote indigenous and exotic vegetables. These vegetables are of great importance in daily diets. Their important nutritional contents including minerals, vitamins, and fibers make them effective in protecting consumers from age-related complications, and from cardiovascular diseases and cancer in general [6,7]. Many plants in general and vegetables in particular are strongly recommended in human nutrition due to their high concentration of antioxidants, their antimicrobial activities and their preventing mercury-related illnesses [8,9]. Among vegetables, leafy ones are the most consumed in Benin. Of all the traditional leafy vegetables,

amaranth (*Amaranthus cruentus*) and Gboma (*Solanum macrocarpon*) are among the most cultivated and consumed [10]. Amaranth leaf yield in Benin is estimated at 30 t/ha [11], which is dependent on the production system. In fact, when a harvest is carried out by eliminating the flowers, the yield can reach up to 32 t/ha against 17.8 t/ha when the floral clusters are not eliminated. A third harvesting system, which consists of pulling the whole amaranth plant out after two cuts, achieves a yield of 29.8 t/ha [12].

Unfortunately, most such vegetables are damaged by numerous plant pathogens among which bacteria are of importance given the severity of their adverse effect upon infection. Recently, bacterial wilt caused by *Ralstonia solanacearum* was reported as a limiting factor in Solanaceae production in Benin [13], and bacterial wilt incidence of 70% on tomato [14] and 75.2% on Gboma [15] were reported. Globally, economic losses due to *R. solanacearum* in potato cultivation were estimated at 950 million USD [16]. In addition to the many host crops from the *Solanaceae*, *R. solanacearum* was reported for the first time in Benin on *Amaranthaceae* crops [13], with 72.4% incidence on amaranth and an equivalent level of yield loss due to entire plant damage.

*R. solanacearum* exhibits an exceptional genotypic, phenotypic, and ecological variability. The pathogen adapts easily to plant species and environments, making its management difficult [17]. DNA analysis of *R. solanacearum* makes it possible to classify the different strains by phylotype, race and biovar. Recently, sequencing of the egl gene has made it possible to distinguish groups of strains within the same biovar called sequevar. This classification method revealed several sequevars including sequevar 56, first reported in China by She et al. [18]. The most effective and environmentally friendly control methods are based mainly on the use of resistant varieties [15]. However, there is no bacterial wilt resistant amaranth variety available in Benin and elsewhere in the world. The present study was aimed at evaluating a collection of amaranth varieties for their resistance to bacterial wilt caused by a highly destructive endemic strain of *R. solanacearum* present in Benin.

## 2. Materials and Methods

### 2.1. Screen House Experiment

**Site and experimental design.** In the screen house, ten varieties of *Amaranthus sp.*, including nine improved (A2002, Bresil (B) -Sel, Madiira 2, AC-NL, GARE ES13-7, Madiira 1, UG-AMES13-2, AM-NKGN, IP-5-Sel) from the World Vegetable Center and one local variety (Fôtètè wéwé) from the PCM Program INRAB in Benin were used. The experiments were carried out under a screen house at the World Vegetable Center's West and Central Africa—Coastal and Humid Regions site located at the IITA-Benin Campus, in the Abomey-Calavi district. This district of southern Benin is bounded at its north side by Zè district, its south side by Atlantic Ocean, its east side by So-Ava and Cotonou districts, and at its west side by Tori-Bossito and Ouidah districts. The average daily temperature was 28 °C with an average relative humidity of 90% during the rainy season. During the dry season, it was 30 °C with a relative humidity of 85%. The experimental design was a randomized complete block with four replications.

**Plant inoculation.** Young plants were grown in pots of 2 L, containing a sterilized mixture of soil and chicken manure (*2 v/1 v*). The plants were inoculated twice. The first inoculation was carried out on 3-week-old seedlings at the time of transplanting after hand cutting the bottom of the roots. The second inoculation was done 2 weeks after transplanting on secondary roots by two cuts on both sides of plants. For this, the roots were scarified using a sterile knife. A suspension of $10^8$ CFU.mL$^{-1}$ was prepared using a known virulent strain LDCAVR M1 of *R. solanacearum* phylotype-I, NCBI GenBank accession N° MH397250. For each inoculation, 40 mL of the bacterial suspension were poured on the collar of each plant. The inoculated plants were watered with 40 mL of tap water twice daily from the day after inoculation. The experimental design was a randomized complete block design (RCBD) with four replications and ten plants per variety for each replication. Control plants underwent the same procedures except that they were drenched with a

40 mL of sterile distilled water on the collar surface and ten plants were used for each replication. The control plants were separated from that of inoculated plants by 4 m. The screen house experiment was repeated for a second time.

**Colonization test.** Four weeks after sowing, all surviving plants were removed from the pots and brought to the laboratory. For each plant, a stem piece of about 10 cm from the collar was cut, washed with tap water, disinfected with 70% ethanol and slightly flamed. Using a sterile scalpel, the basal cut was refreshed. The cup section was used to place five imprints on modified semiselective medium of South Africa (SMSA) in petri dishes [19], and incubated for 48 h at 30 °C.

*2.2. Field Experiment*

Two different experiments were carried out on the WorldVeg experimental site at the IITA-Benin station: during the rainy season from May to July 2018 and the dry season from November 2018 to January 2019. In the rainy season, the average daily temperature was 27 °C with an average relative humidity of 85%, whereas in the dry season the average temperature was 28 °C with a relative humidity of 84%. The experiment consisted of testing of the 10 varieties, which were used in the screen house trial in a field with natural infection of *R. solanacearum*. The bacterial strain used for inoculation in screen house was the same strain isolated from this field site. Under field conditions, the experiment was conducted in an RCBD with three replications. In each plot, 78 plants were transplanted in six rows at a rate of 13 plants per row over an area of 3.6 m$^2$. Data were collected on the four central rows. From the four central rows, plants of two rows were harvested by cutting 28 days after sowing.

**Data collection**. In the field, wilted plants were collected before and after cutting the two rows of plants, and during the cycle for plants on the left rows. The knives were sanitized after their use in each plot by flaming them using the 90° alcohol. Two observations were made per week, yielding a total of 16 observations. In the screen house, data collection was carried out by assessing disease symptoms at 2-day intervals until 28 days after inoculation (DAI) using the disease scale of:

- 0 = no symptoms (no wilt)
- 1 = plant three quarter wilted, completely wilted or plant dead. From these data, the following parameters were determined.

**Incidence of bacterial wilt (IBW): Proportion of symptomatic plants**.
**Bacterial colonization index (BCI).**

Proportion of infected amaranth plants. It was determined at 28 DAI as follows: BCI = (number of wilted plants + number of colonized)/total number of plants.

**Area under the disease progress curve (AUDPC).** The AUDPC was determined for both the field and screen house data using the incidence of bacterial wilt at each collection using the following formula [20,21]:

$$AUDPC = \sum_{i=1}^{k}[(IBW_i + IBW_{i+1})(t_{i+1} - t_i)]/2$$

With i = the collection period and $t_i$ = the date of the collection period.

Based on the IBW and AUDPC, amaranth varieties were classified into three groups as noted in Table 1.

**Table 1.** Resistance classification of amaranth varieties used in this study.

| Resistance Class | IBW (%) | AUDPC (%-Days) |
| --- | --- | --- |
| Resistant | 0–15 | 0–200 |
| Moderately resistant | 15–40 | 200–400 |
| Susceptible | >40 | >400 |

*2.3. Statistical Analysis*

Mixed-model analysis of variance (ANOVA) with amaranth varieties and measurement times as fixed factors and blocks as a random factor was performed to assess the effect of different factors on IBW. The AUDPC was determined for each amaranth variety. The analysis was carried out in the R version 3.5. 3 statistical environment [22]. The Student–Newman–Keuls test at the $p = 0.05$ was used to discriminate the means into homogeneous groups [23]. Pearson's correlation test was performed to relate the calculated bacterial wilt incidences in the screen house and in the field.

## 3. Results

*3.1. Screening of Amaranth Varieties for Resistance to Bacterial Wilt in the Screen House*

The results revealed differences in term of diseases expression between varieties, over time and their interaction for experiment 1 ($p < 0.001$) and experiment 2 ($p < 0.003$) (Table 2). For both experiments, none of the non-inoculated control plants wilted during the trial (data not shown). The analysis of variance carried out on AUDPC showed highly significant differences between varieties in experiment 1 ($p < 0.001$) and in the experiment 2 ($p = 0.004$) (Table 3). AUDPC was higher in experiment 2 than in experiment 1 for all varieties except the varieties AC-NL, Madiira 1 and Madiira 2. AUDPC was high for IP-5-Sel, GARE ES13-7, Bresil (B)-Sel, AC-NL, Madiira 1 and A2002, moderately high for the varieties Madiira 2, AM-NKGN Benin-local-variety and low for UG-AMES13-2 (Table 3).

**Table 2.** Effect of variety and period of measurement on the wilting incidence in the screen house.

| Source of Variation | DF | Experiment 1 | | Experiment 2 | |
|---|---|---|---|---|---|
| | | F | P | F | P |
| Varieties | 9 | 11.33 | 0.000 | 5.04 | 0.002 |
| Time | 3 | 204.00 | 0.000 | 246.90 | 0.000 |
| Varieties × Time | 27 | 4.65 | 0.000 | 3.18 | 0.000 |

DF = degree of freedom; F = Fisher value; $p$ = probability.

**Table 3.** Area under disease progress curve (AUDPC) of bacterial wilt incidence in the screen house.

| Varieties | AUDPC (%-Days) | |
|---|---|---|
| | Experiment 1 | Experiment 2 |
| AC-NL | 711.7 [a] | 583.3 [ab] |
| IP-5-Sel | 665.0 [a] | 1131.7 [a] |
| GARE ES13-7 | 653.3 [a] | 840.0 [ab] |
| Bresil (B)-Sel | 466.7 [ab] | 700.0 [ab] |
| Madiira 1 | 455.0 [ab] | 350.0 [b] |
| Madiira 2 | 256.7 [bc] | 198.3 [b] |
| AM-NKGN | 163.3 [bc] | 233.3 [b] |
| Benin-local-variety | 116.7 [bc] | 338.3 [b] |
| A2002 | 81.7 [bc] | 431.7 [b] |
| UG-AMES13-2 | 0.0 [c] | 163.3 [b] |
| $p$ | <0.001 | <0.001 |

Mean value(s) followed by different letter(s) in a column are significantly different at 5%.

*3.2. Bacterial Wilt Incidence and Bacterial Colonization Index*

The average incidence of bacterial wilt varied between varieties for the two seasons of experiments (Figures 1 and 2). It was very high for the IP-5-Sel, AC-NL, GARE ES13–7, Bresil (B) -Sel, Madiira 1 and A2002 varieties, ranging from 76.7% to 100%, moderate for the AM varieties -NKGN, Madiira 2 and Benin-local-variety with 30% to 46% and very low for the UG-AMES13–2 variety with 0% and 13.3% in experiment 1 and experiment 2, respectively (Figures 1 and 2). In experiment 1, the average bacterial colonization index was high for all amaranth varieties with 100% colonization except for varieties UG-AMES13–2

and Madiira 2 (Figure 1). UG-AMES 13–2 did not show any incidence of bacterial wilt, but was latently-infected. In experiment 2, all amaranth varieties tested were 100% colonized by *R. solanacearum* (Figure 2).

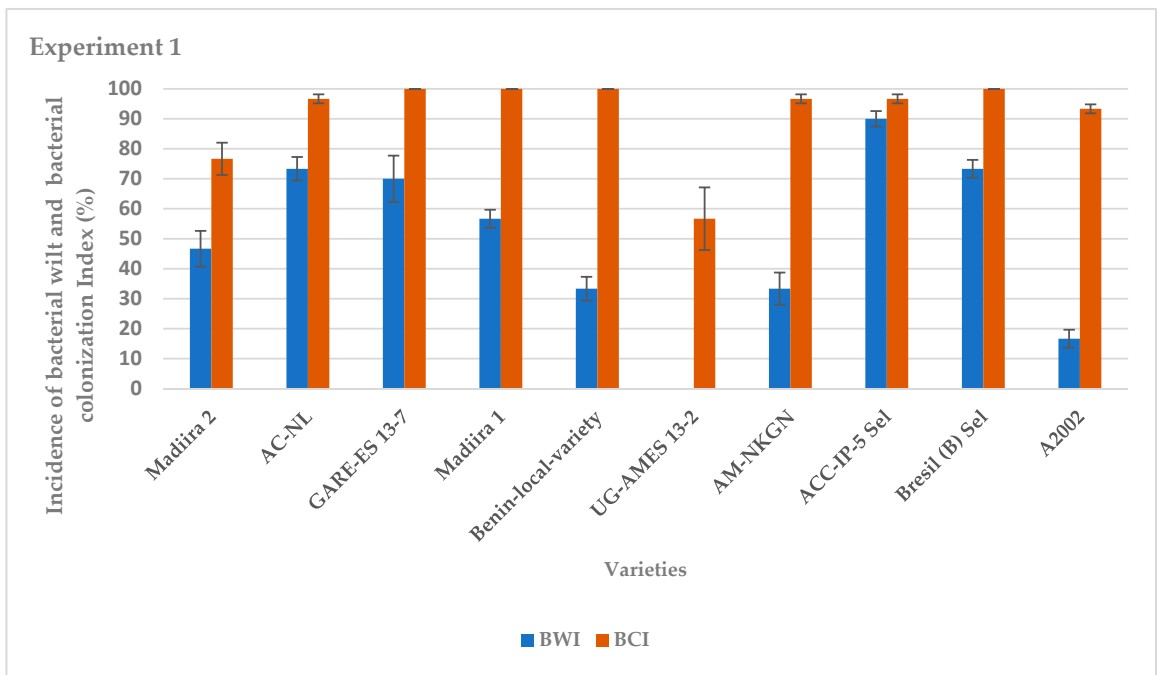

**Figure 1.** Incidence of bacterial wilt (IBW) and bacterial colonization index (BCI) during experiment 1 in the screen house. Error bars indicate the standard error of mean (S.E.M).

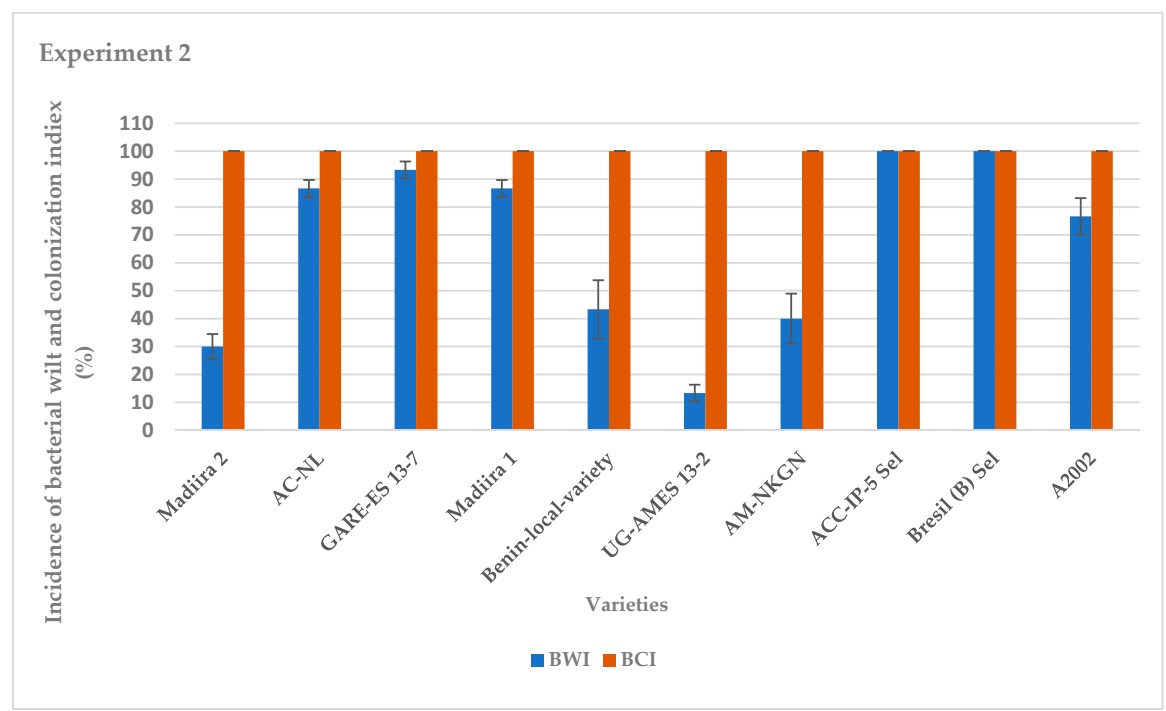

**Figure 2.** Incidence of bacterial wilt (IBW) and bacterial colonization index (BCI) during experiment 2 in the screen house. Error bars indicate the standard error of mean (S.E.M).

### 3.3. Resistance Classes of Amaranth Varieties

The results revealed three resistance groups among amaranth varieties tested (Table 4). UG-AMES13-2 was classified as resistant, Madiira 2, AM NKGN and Benin local variety as moderately resistant, and others as susceptible. According to the defined resistance classes, the amaranth varieties IP-5-Sel, AC-NL, GARE ES13-7, Bresil (B) -Sel, Madiira 1 and A2002 were identified as susceptible to bacterial wilt.

**Table 4.** Ranking of ten amaranth varieties according to bacterial wilt incidence (BWI) and the area under disease progress curve (AUDPC) based on screen house results.

| Varieties | Mean BWI (%) | Mean AUDPC (%-Days) | Resistance Class [a] |
|---|---|---|---|
| IP-5-Sel | 95.0 | 1291.1 | S |
| AC-NL | 80.0 | 1156.6 | S |
| GARE ES13-7 | 81.7 | 1107.4 | S |
| Bresil (B)-Sel | 86.7 | 679.1 | S |
| Madiira 1 | 71.7 | 674.9 | S |
| A2002 | 46.7 | 455.0 | S |
| Madiira 2 | 38.3 | 227.5 | MR |
| AM-NKGN | 36.7 | 215.7 | MR |
| Benin-local-variety | 38.3 | 311.4 | MR |
| UG-AMES13-2 | 6.5 | 167.8 | R |

[a] S = susceptible; MR = moderately resistant; R = resistant.

### 3.4. Field Resistance of Uncut Plants of Amaranth Varieties

ANOVA of BWI showed a very high significant difference between varieties, time and their interaction in both experiments, except for the variety and time interaction in experiment 2 (Table 5). ANOVA of AUDPC also revealed very high significant differences between amaranth varieties in experiment 1 ($p = 0.000$) and in experiment 2 ($p = 0.007$) (Table 6). In experiment 1, the incidence of bacterial wilt was very high for varieties IP-5-Sel and AC-NL, moderately high for varieties GARE ES13-7 and Madiira 1, and low for varieties A2002, Bresil (B)-Sel, Madiira 2, AM-NKGN, Benin-local-variety and UG-AMES13-2. The AUDPC mean values in experiment 2 were generally higher compared to those in experiment 1. The recorded AUDPC values were very high for varieties IP-5-Sel, AC-NL, GARE ES13-7, Madiira 1 and Bresil (B)-Sel, moderate for A2002, Madiira 2 and Benin-local-variety and low for AM-NKGN and UG-AMES (Table 6).

**Table 5.** Effect of variety, time and interaction on the incidence of bacterial wilt on amaranth plants before cutting in the field.

| Source of Variation | DF | Experiment 1 | | Experiment 2 | |
|---|---|---|---|---|---|
| | | F | P | F | P |
| Varieties | 9 | 6.65 | 0.000 | 4.46 | 0.003 |
| Time | 3 | 23.67 | 0.001 | 62.00 | 0.000 |
| Varieties × Time | 27 | 2.03 | 0.014 | 1.26 | 0.230 |

### 3.5. Field Resistance of Cut Plants of Amaranth Varieties

Significant effects of varieties, time, and interaction were noted (Table 7). The analysis revealed highly significant difference between amaranth varieties for the two experiments ($p < 0.001$). AUDPC values were higher in experiment 2 compared to experiment 1 for all varieties tested (Table 8). The area under the disease progress curve was high for the varieties IP-5-Sel, AC-NL, GARE ES13-7, Madiira 1, Bresil (B)-Sel, moderate for the varieties A2002, Madiira 2 and Benin -local-variety and low for varieties AM-NKGN and UG-AMES13-2 in both the experiments.

**Table 6.** Area under disease progress curve (AUDPC) for uncut amaranth plants in the field.

| Varieties | AUDPC (%-Days) | |
|---|---|---|
| | Experiment 1 | Experiment 2 |
| IP-5-Sel | 1426.9 [a*] | 1557.1 [a*] |
| AC-NL | 1180.1 [a] | 1637.8 [a] |
| GARE ES13-7 | 596.8 [b] | 1817.3 [a] |
| Madiira 1 | 471.2 [b] | 1041.0 [ab] |
| A2002 | 152.6 [b] | 753.9 [ab] |
| Bresil (B)-Sel | 260.3 [b] | 1076.9 [ab] |
| Madiira 2 | 336.5 [b] | 879.5 [ab] |
| AM-NKGN | 80.8 [b] | 323.1 [ab] |
| Benin-local-variety | 112.2 [b] | 587.8 [ab] |
| UG-AMES13-2 | 13.5 [b] | 394.8 [b] |
| *p* | 0.000 | 0.007 |

* Mean value(s) followed by different letter(s) in a column are significantly different at 5%.

**Table 7.** Effect of variety, time and interaction on the incidence of bacterial wilt on cut plants of amaranth in the field.

| Source of Variation | DF | Experiment 1 | | Experiment 2 | |
|---|---|---|---|---|---|
| | | F | | F | P |
| Varieties | 9 | 6.21 | 0.001 | 12.94 | <0.001 |
| Time | 3 | 34.52 | <0.001 | 215.20 | <0.001 |
| Varieties × Time | 27 | 4.63 | <0.001 | 2.02 | 0.014 |

**Table 8.** Area under the disease progress curve (AUDPC) for bacterial wilt incidence on cut plants of amaranth varieties in the field.

| Varieties | AUDPC (%-Days) | |
|---|---|---|
| | Experiment 1 | Experiment 2 |
| IP-5-Sel | 1409.0 [a] | 1557.1 [a] |
| AC-NL | 1189.1 [ab] | 1637.8 [a] |
| GARE ES13-7 | 919.9 [abc] | 1817.3 [a] |
| Madiira 1 | 691.0 [bcd] | 1041.0 [b] |
| A2002 | 556.4 [bcd] | 753.9 [bc] |
| Bresil (B)-Sel | 493.6 [bcd] | 1076.9 [b] |
| Madiira 2 | 448.7 [bcd] | 879.5 [bc] |
| AM-NKGN | 170.5 [cd] | 323.1 [c] |
| Benin-local-variety | 125.6 [cd] | 587.8 [bc] |
| UG-AMES13-2 | 40.4 [d] | 394.9 [c] |
| *p* | <0.001 | <0.001 |

Mean value(s) followed by different letter(s) in a column are significantly different at 5%.

### 3.6. Correlation between Screen House and Field Disease Incidence

Pearson's test revealed a positive and significant relationship at the 5% level between the incidence of bacterial wilt in the screen house and in the field for cut and uncut plants during the two experiments (Table 9).

**Table 9.** Correlation coefficient between the incidence of bacterial wilt in screen house and field experiments.

| Incidence of Bacterial Wilt | Correlation Coefficient | |
|---|---|---|
| | **Experiment 1** | **Experiment 2** |
| Screen house x Field with cut plants | 0.7454 * | 0.5680 * |
| Screen house × Field with uncut plants | 0.6102 * | 0.5541 * |

* significance at 5%.

## 4. Discussion

The field and screen house experiments were carried out in both the dry and rainy seasons. For all varieties, the results showed that the disease was less expressed during the rainy season compared to the dry season. Thus, amaranth seemed to be more susceptible to bacterial wilt in the dry season with higher daily temperatures than in the rainy season. Our results are similar to that of Techawongstien et al. [24], who reported that bacterial wilt was more severe in the dry season for tomato plants cultivated on contaminated soils than in the rainy season. Similarly, Singh et al. [25] reported the high susceptibility of two varieties (one highly susceptible and one moderately resistant) after inoculation of their strain of *R. solanacearum* and incubation of the plants at 30 °C. In this study no wilting was recorded at 20 °C. In Taiwan, a comparative study of three strains of *R. solanacearum* showed in tomato and potato crops that the strains were more virulent at 24 °C to 28 °C than at 20 °C with less severity at 24 °C than 28 °C [26]. In tobacco production, an increased susceptibility of five cultivars of tobacco inoculated with a strain of *R. solanacearum* was reported at 30 °C and 35 °C while no wilt was observed up to 15 °C incubation temperature [27]. Based on these results, we conclude that the high temperature increases infection of *R. solanacearum* strains and decreases resistance in case of amaranth varieties under the conditions prevalent in Benin. This may be due to the fact that in the dry season, plants absorb more inoculum of *R. solanacearum* with water from the soil. These bacteria, once in the plants, grow and quickly cause infection, but this hypothesis needs to be further studied.

Most of the varieties showed significantly higher disease incidence and area under disease progress curve on cut plants than on uncut ones. Zocli [15] reported similar observations on bacterial wilt dissemination in Gboma (*S. macrocarpon*) plants. It was found that bacterial wilt incidence of *S. macrocarpon* was low before cutting plants and very high after cutting. According to the author, infection/penetration could be favored by the openings left by the wounds on stem while cutting.

The results revealed diversity in resistance/susceptibility to *R. solanacearum* infection among amaranth varieties. These results confirm those of Sikirou et al. [13] who first reported amaranth as a host plant for *R. solanacearum*. Diversity in reaction of cultivars of tomato and *S. macrocarpon* to *R. solanacearum* infection has been reported by Oussou et al. [28]. In the present study only the variety UG-AMES13-2 (an *Amaranthus dubius* variety) over ten amaranth varieties tested was found to be resistant to *R. solanacearum* strain MH397250/NCBI 5 used. This variety may have gene/s responsible for bacterial wilt resistance, which needs to be confirmed in future studies. Sources of resistance to pathogens in many other plants have been well documented [29–31], but not in amaranth. Thus, the susceptibility of IP-5-Sel, AC-NL, GARES-ES 13-7, Bresil (B) -Sel, Madiira 1 and A2002 to bacterial wilt caused by *R. solanacearum* could be related to the absence of resistance gene/s in their genome, which requires further investigation.

The results revealed that all the tested varieties were colonized by the bacteria regardless of their degree of resistance to the disease. *R. solanacearum* may colonize resistant amaranth varieties in the same way as the susceptible and moderately susceptible varieties. Colonization of a wide range of resistant species and genotypes to bacterial wilt has already been reported on potato [32], tomato [33] and African nightshade [15]. In addition, it will be important to conduct histological studies to identify differences in bacterial localization and spread among resistant, moderately resistance and susceptible varieties. The lower

bacterial colonization index on the two varieties noted may suggest such a scenario. Similar observations have been previously reported for tomato varieties [34].

Across seasons and cropping systems, the reaction of amaranth varieties to bacterial wilt in the field was not different from that of screen house. The AUDPC from both field and screen house experiments ranked the varieties in the same order of susceptible/resistant groups. These results show that the natural inoculum in the soil of the experimental site was well distributed, and sufficient to induce infection of the amaranth plants. Thus, for *R. solanacearum*, selection of varieties for resistance can be based on the results from well contaminated soil. Further studies using various strains of the bacterium affecting amaranth will be needed to understand the broad utility of the resistance over larger geographical landscapes.

**5. Conclusions**

The present study revealed more susceptibility of amaranth varieties to *R. solanacearum* infection in the dry season and when the plants were harvested by stem cutting. Bacterial wilt incidence after artificial inoculation did not differ from that of the naturally contaminated soil in the field. All varieties tested were heavily colonized by the bacteria. However, out of the ten varieties tested, six were identified as susceptible, three as moderately-resistant and one as resistant to bacterial wilt caused by *R. solanacearum*. Thus, the resistant amaranth variety UG-AMES13-2 can be considered as the first choice against *R. solanacearum* in Benin. It is also recommended to the breeders to identify and introduce their resistance characteristics into breeding lines with the goal of developing more resistant varieties commercially for farmers in Benin and elsewhere.

**Author Contributions:** Conceptualization, R.S. (Rachidatou Sikirou), M.E.D. and J.H.; methodology, R.S. (Rachidatou Sikirou), M.E.D. and J.H.; software, R.S. (Rachidatou Sikirou) and M.E.D.; validation, R.S. (Rachidatou Sikirou), M.P., W.B., V.A.-S., R.S. (Ramasamy Srinivasan); investigation, R.S. (Rachidatou Sikirou), M.E.D. and J.H.; resources, R.S. (Rachidatou Sikirou), V.A.-S.; data curation, M.E.D., J.H.; writing—original draft preparation, R.S. (Rachidatou Sikirou), M.E.D. and J.H.; writing—review and editing, R.S. (Rachidatou Sikirou), M.P., W.B., V.A.-S., R.S. (Ramasamy Srinivasan); supervision, R.S. (Rachidatou Sikirou); funding acquisition, R.S. (Ramasamy Srinivasan). All authors have read and agreed to the published version of the manuscript.

**Funding:** Funding for this research was provided by the World Vegetable Center and by long-term strategic donors to the World Vegetable Center: Taiwan Government, the Foreign, Commonwealth & Development Office (FCDO) from the UK government, United States Agency for International Development (USAID), Australian Centre for International Agricultural Research (ACIAR), Germany, Thailand, Philippines, Korea and Japan and The Conservation Food and Health Foundation to the University of Florida, USA and INRAB, Benin.

**Institutional Review Board Statement:** Not applicable.

**Informed Consent Statement:** Not applicable.

**Data Availability Statement:** The data presented in this research are available from the corresponding author upon reasonable request.

**Acknowledgments:** The authors are grateful to Aubin Amagnide for data analysis and Frejus Medetonvoh for technical assistance. We are also thankful to Azoma Komla for helping in the field trial implementation and data collection.

**Conflicts of Interest:** The authors declare no conflict of interest.

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
