# Peer review of "Screening of Amaranthus sp. Varieties for Resistance to Bacterial Wilt Caused by Ralstonia solanacearum"

_horticulturae, doi:10.3390/horticulturae7110465_

Round 1

Reviewer 1 Report

This publication describes the important problem of Ralstonia infections in plants. Ralstonia is a bacterium that is eradicated ex officio in many countries and cases of Ralstonia infection must be reported and treated immediately. The search for plant varieties resistant to Ralstonia is very important and may lead to at least a partial reduction of the bacterium in crops.
The study was properly designed. I have no remarks on the merits of the work. 

Few editorial remarks:

Line 37: Not sufficient repetition

Lines 72-74: Please rewrite this sentence to be more clear

I would also recommend changing the graphs to colorful ones, in order to make the work more varied and attractive. 

Reviewer 2 Report

In the manuscript titled “Screening of Amaranthus sp. varieties for resistance to bacterial 3 wilt caused by Ralstonia solanacearum” the authors investigate a collection of Amaranth varieties for their resistance to bacterial wilt caused by a highly destructive endemic strain of R. solanacearum present in Benin.

I thing that this work was carefully conducted. The paper is well written but in my opinion it lacks just some aspects. It should be integrated considering the following aspects:

  • check throughout the manuscript that mL is written instead of ml
  • the abstract is not very clear and there are too many acronyms
  • the experimental plant inoculation plan because it was done that way?
  • Better explain controls in experimental procedures
  • Better explain the reasons for the different response of the different varieties in particular the resistant, the moderately resistant, and the susceptible
  • I suggest extending the introduction or the discussion describing that some plant present

Antimicrobial and antioxidant activity in the proteins extracted from the fruit. For this aim I suggest to read and quote the following work:

Antimicrobial and antioxidant activity of proteins from Feijoa sellowiana Berg. fruit before and after in vitro gastrointestinal digestion

DOI: 10.1080/14786419.2018.1543686

  • In some cases the same extract are able to Protect Human Red Blood Cells from Mercury-Induced Cellular Toxicity

For this aim aim I suggest to read and quote the following work:

Phenol-Rich Feijoa sellowiana (Pineapple Guava) Extracts Protect Human Red Blood Cells from Mercury-Induced Cellular Toxicity

DOI: 10.3390/antiox8070220

Perhaps these papers can help better argue the different levels of resistance found among the varieties studied

Round 2

Reviewer 2 Report

Accept in present form

Author Response

Dear Reviewer

Please find attached the response to your suggestion.

Best Regards
